# Flexural Properties and Mean Intrinsic Flexural Strength of Old Newspaper Reinforced Polypropylene Composites

**DOI:** 10.3390/polym11081244

**Published:** 2019-07-26

**Authors:** Quim Tarrés, Jordi Soler, José Ignacio Rojas-Sola, Helena Oliver-Ortega, Fernando Julián, F. Xavier Espinach, Pere Mutjé, Marc Delgado-Aguilar

**Affiliations:** 1Laboratory of Paper Engineering and Polymer Materials, Department of Chemical Engineering, University of Girona, 17003 Girona, Spain; 2Department of Architecture and Construction Engineering, University of Girona, 17003 Girona, Spain; 3Department of Engineering Graphics, Design and Projects, University of Jaen, 23071 Jaen, Spain; 4Design, Development and Product Innovation, Department Of Organization, Business Management and Product Design, University of Girona, 17003 Girona, Spain

**Keywords:** natural fibers, recycled paper, composites, polypropylene, flexural strength

## Abstract

Newspapers have a limited lifespan, and therefore represent a focus of used and disposed paper. While these refuses have a considerable value and can be easily recycled, a considerable fraction ends in landfill. The authors show the possibility of adding value to used newspaper and enlarge its value chain. Old newspaper incorporates a high amount of wood fibers able to be used as reinforcement. On the other hand, this material also incorporates inks and other components inherent to newspaper production. In this work, the authors disintegrate old newspaper to recover and individualize wood fibers. A morphological analysis showed that the recovered fibers had aspect ratios higher than 10, revealing, a priori, their strengthening capabilities. Thereupon, these fibers were compounded with polypropylene at different contents, ranging from 20% to 50% *w*/*w*. It is well known that wood fibers are hydrophilic, while polyolefin are hydrophobic. This is a drawback to obtaining strong interfaces. Thus, two sets of composites were produced, with and without a coupling agent. The results showed that uncoupled composites increased the flexural strength of the matrix but reached an equilibrium point from which adding more reinforcement did not changer the flexural strength. On the other hand, the coupled composites showed a linear increase of the flexural strength against the reinforcement content. The flexural moduli of the coupled and uncoupled composites were very similar and evolved linearly with the reinforcement content.

## 1. Introduction

Newspapers are products with very low lifespan. Nonetheless, they are widely produced, consumed and discarded [1]. Under ideal conditions, old newspaper are prone to be recycled as pulps but the reality is that a high percentage are dumped, being one of the most collected wastes [2]. Waste paper recycling has increased, mainly due to the society environmental concern, the diminution of the forested areas and the regulations in support of its recycling. Old policies based on dumping or incinerating are decreasing in importance due to the increasing costs of landfilling and the concerns towards CO_2_ emissions [1,3,4]. Thus, in addition to its recovery as paper pulp, other uses for waste paper can be explored.

The use of fibers from wood or agroforestry as replacement of glass fiber has been explored as a way to formulate and produce greener materials [5,6,7]. Nonetheless, there are few studies devoted to the use of recovered newspaper as a source of reinforcing fibers. Anyhow, these studies reveal promising mechanical properties for such composites [7,8,9,10]. The literature studies mainly the tensile properties and the strength of the interphase between old newspaper fibers (ONPF) and the matrices [8,9,11,12,13]. A high amount researches are devoted to the use of coupling agents as maleated polyolefin to increase the strength of the interphases [13,14,15,16]. This has been a widely treated aspect for natural fibers reinforced composites [17,18,19,20,21]. The different natures of the phases, being natural fibers hydrophilic and polyolefin hydrophobic prevents the creation of interactions between them, resulting in weak interphases [22,23]. While different fiber treatments and coupling agents have been tested, the use of maleated polyolefin has rendered the best results [22,24,25].

The literature shows how ONPF reinforced polypropylene can replace a glass fiber reinforced polypropylene for products like a pool water pump body [1]. Nonetheless, the study was based on the tensile properties of the composites, ignoring the flexural properties of such materials. These flexural properties, to the best knowledge of the authors, have been scantly reported. Injection molded short fiber reinforced composites, tend to show a semi-aligned orientation of the fibers. The mean orientation means that the composites will be anisotropic and their tensile strength will be orientation dependent [23,26]. In these cases, the flexural properties of the materials are of great interest to the engineers.

In this paper, composite materials based on ONPF reinforced polypropylene were formulated, prepared and tested under flexural loads. The composites were reinforced with 20% to 50% *w*/*w* of ONPF. Coupled and uncoupled sets were prepared. The coupled composites added 6% of maleated polypropylene. The materials were mold injected to obtain standard specimens. The specimens were tested under three-point bending conditions and the results were discussed and compared with other materials. The micromechanics of the flexural strength was investigated by using a fiber flexural strength factor. The intrinsic flexural strength of ONPF was computed from a relation between the strength factors for the tensile and flexural modes. Finally, a modified rule of mixtures was used to obtain the intrinsic flexural strength of the reinforcements.

## 2. Materials and Methods

### 2.1. Materials

The matrix was a polypropylene homopolymer Isplen PP090 (Repsol-YPF, Tarragona, Spain).

Rotimprès (Girona, Spain) kindly provided samples of used Punt Diari newspaper. The thermomechanical pulp used to obtain the newspaper contained 85% of thermomechanical pulp from hardwood and 15% of calcium carbonate, in addition to inks.

The density of the matrix and the fibers were measured at 0.906 and 1350 g/cm^3^, respectively.

Epolene G305 by Eastman Chemical Products (Barcelona, Spain) was used as coupling agent. It is a polypropylene functionalized with maleic anhydride (MAPP).

### 2.2. Obtention of the Reinforcing Fibers

Old newspapers were cut to 10 by 10 mm portions and introduced in a water bath with 1% of NaOH. The mix was disintegrated in a laboratory pulper Pucel by Metrotec (Lezo, Spain) with an effective volume of 20 L. The process was carried out at 50 °C, 20 rev/s rotor speed and 10% consistency. The resulting pulp was filtered and dried in an oven at 80 °C. During this process, a 5% of the CaCO_3_ was lost.

In order to ensure a correct dispersion of the ONPF in the matrix, these fibers were dispersed in a (1:3) water–diglyme solution. The use of diglyme has been reported to increase the quality of the dispersion of the fibers by avoiding the creation of bonds between the cellulosic fibers [8,11]. Figure 1 shows a flowchart of the experimental procedure.

### 2.3. Composite Preparation and Sample Obtaining

The composite materials were formulated with 20%, 30%, 40% and 50% *w*/*w* ONPF contents. The composites were mixed in a Brabender^®^ plastograph internal mixing machine. The mixer was operated at 180 °C and 80 rpm during 10 min. Coupled composites added 6% *w*/*w* of MAPP with respect to the reinforcement content. The composites were pelletized in a blades mill and stored in an oven at 80 °C during 24 h.

Composite materials will be referred with the nomenclature x%ONPF–y%MAPP, where x and y are the ONPF and MAPP percentages *w*/*w*, respectively.

The samples for the three-point bending test were produced with a steel mold in an injection-molding machine (Meteor 40, Mateu and Solé, Barcelona, Spain). At least ten specimens were obtained for the different formulations.

### 2.4. Mechanical Characterization

Composite specimens were pre-conditioned according to ISO D618 protocol standard. Flexural tests were carried out with an Instron 1122 universal testing machine (Instron, Norwood, United States), following ASTM F790 regulation. At least five specimens of all the composite formulations were tested.

The strain under bending of the composites (*ε*_f_*^C^*) was computed according to Equation (1):(1)$$\varepsilon_{f}^{C}=\frac{6\times D\times d}{L^{2}}.$$
where *D* and *d* are the deflection at the center of the specimen and its thickness, respectively. *L* is the supports span.

### 2.5. Morphology of the Fibers

ONPF were recovered by matrix solubization using a Soxhlet apparatus. The solvent was decalin [11]. The obtained fibers were rinsed with acetone and distilled water and dried in an oven, at 105 °C during 24 h.

Then, the ONPF were morphologically characterized in a Kajaani analyzer (FS-300) (Valmet, Keilasatama, Findland) [1,11]. The equipment returned the fiber length and width distributions, as well as mean and weighted lengths.

## 3. Results and Discussion

### 3.1. Flexural Properties of the Composites

The tensile strength was highly impacted by the presence of coupling agents [1,11]. This was also observed for the flexural strength. Table 1 shows the ONPF volume fractions (*V^F^*) and the obtained flexural strength (*σ*_f_*^C^*), flexural modulus (*E*_f_*^C^*), strain at break (*ε*_f_*^C^*) and contribution of the matrix to the flexural strength of the composite (*σ*_f_*^m*^*), and their standard deviations.

The coupled and uncoupled composites behaved differently against ONPF contents. While uncoupled composites showed an initial increase on the flexural strength (33%), compared to the matrix, this increase remained almost constant (around 57%), for ONPF percentages higher than 30% *w*/*w*. Besides, the 50%ONPF–0%MAPP composite showed a flexural strength 50% higher than the matrix. On the other hand, coupled composites showed regular increases of their flexural strength against ONPF contents. The composites with ONPF contents ranging from 20 to 50% *w*/*w* showed 59%, 90%, 97% and 116% increases compared to the flexural strength of the matrix.

The flexural moduli and strains at break are presented as additional information as has been discussed in a prior research [27]. Nonetheless, it was observed how, despite the presence of coupling agents, the flexural modulus increased similarly for the coupled and uncoupled materials. On the other hand, the strain at break of the coupled composites was found to be higher than the uncoupled, being a direct consequence of Hooke’s law.

In the case of the coupled composites, their flexural strength showed a linear evolution against ONPF content that can be fitted to a regression line with a correlation coefficient (*r*^2^) of 0.96. This behavior has been reported as an indication of a good dispersion as well as a strong interphase [6,28,29]. The hypothesis of a good dispersion is supported by the linear evolution of the flexural moduli against ONPF contents, but needs more corroboration [10]. On the other hand, the erratic evolution of the flexural strength of uncoupled composites is an indication of a weak interphase. In any case, obtaining strong interphases involves: good dispersion, without fiber bundles creation, mechanical anchoring, due to a good wetting of the fibers and the absence of voids, and the creation of chemical interactions between the phases [22].

The surface of natural fibers is mainly composed of cellulose, hemicellulose and lignin, with lesser amounts of extractives [30,31]. This composition changes noticeably when the fibers are treated. These treatments are used in the papermaking industry and mostly eliminate part of the lignin and the extractives. The elimination of the lignin bleaches the fibers and increases the presence of cellulose and hemicellulos in the surfaces of the fibers. The presence of higher amounts of cellulose results on a higher amount of hydroxyl groups, prone to create hydrogen bonds. On the one hand this is positive because increases the possible chemical interactions with the matrix, but in the other hand also increases the interactions between the fibers that tend to create bonds, interlock and difficult its dispersion inside the matrix. In this case the ONPF were dispersed in diglyme to avoid the creation of bonds between the ONPF and enable its dispersion in the composite [32].

The presence of voids in the interphase is mainly due to the chemical nature of the phases. Natural fibers are hydrophilic and Polypropylene in hydrophobic, hindering a good compatibility between those phases [23]. Figure 2 shows the interphase of coupled and uncoupled composites.

The uncoupled composite (Figure 1a) shows a clear separation between the fibers and the matrix. This separation is regular all around the visible frontier between the fiber and the matrix, clearly hindering a correct wetting of the fibers and the creation of strong mechanical anchorages. On the other hand, the absence of contact inhibits the creation of chemical bonds. Thus, the increase of the flexural strength is probably governed by weak mechanical anchorages. There is also a noticeable void due to ONPF slippage. The walls of this void are smooth, without any remain of the fiber surface. The interphase is visually very different in the case of the coupled composite (Figure 1b). There are no voids or separations in the visible frontiers between the phases. Additionally, the surface of the fiber is rough, and at its end can be observed some remains of adhered matrix. Thus, it is clear that adding the coupling agent enabled the strength of the interphase.

Figure 3 shows the theoretical mechanisms that take place in the interphase.

ONPF like other lignocellulosic fibers, due to the presence of cellulose and hemicellulose in its surface dispose of a quantity of hydroxyl groups. These groups are unable to interact chemically with the PP. The use of MAPP enables stablishing coupling mechanisms between the maleic acid and the OH groups based on the creation of hydrogen bonds and ester linkages due to the chemical reactions between MAPP anhydride groups and the hydroxyl groups of the fibers [33]. On the other hand, the PP chains of the MAPP will entangle with the matrix [5].

### 3.2. Competitiveness of ONPF Reinforced PP Composites

In order to study the competitiveness of ONPF-based composites, its flexural strength will be compared with a stone groundwood (SGW) lignocellulosic fiber reinforced polypropylene and with a glass fiber reinforced PP. Comparing with another natural fiber reinforced composite serves to identify advantages or disadvantages of ONPF in front of a similar reinforcement. The main difference is that SGW is a raw material provided from a mechanical treatment, and ONPF are recycled thermo–mechanical fibers.

SGW-based composites also needed the presence of MAPP to obtain strong interphases and exploit the strengthening abilities of the reinforcements [30,31]. Figure 4 shows a comparison between the flexural strengths of ONPF and SGW reinforced PP composites.

Uncoupled composites showed a similar behavior; with initial increases of the flexural strength when 20% *w*/*w* of reinforcement was added to the composite. Adding higher percentages of reinforcement only increased slightly the flexural strength. Anyhow, there was no correlation between reinforcement content and flexural strength [34]. Initially the ONPF-based composites showed higher flexural strengths, but at 50% *w*/*w* contents, the value was virtually the same. The higher flexural strength in the case of the ONPF-based composites can be due to the use of diglyme that possibly allowed a better dispersion of such reinforcements. SGW is a mechanical pulp, and the lignin contents allow for a good dispersion without using any reagent.

The case of the coupled composites is also similar. Both, ONPF and SGW-based composites increased their flexural strengths linearly against reinforcement contents. The impact of the coupling agent is clear. Composites reinforced with ONPF and SGW showed similar flexural strengths, at the exception of the materials with 50% *w*/*w* reinforcement contents. In this case, the SGW-based composites reached a higher flexural strength. The cause can be due to the grade of dispersion or to the morphology of the fibers. Having in account the linear behavior of the flexural modulus, the cause can be attributed to the morphologic transformations of the fibers during the composite preparation processes. It is accepted that during these processes the fibers tend to shorten due to attrition phenomena. The fiber shortage increases with the reinforcement content. Additionally, while SGW is a mechanical pulp, ONPF is a raw thermomechanical pulp, recovered and defibrated by a mechanical treatment. Thus, ONPF has suffered two treatments.

The most used treatments are mechanical (MP), thermomechanical (TMP) and chemo–thermomechanical (CTMP). A mechanical treatment involves a defibering under cold aqueous conditions in a Sprout–Waldron or a similar equipment. This treatment has a yield of almost the 100% with respect to the raw material. A thermomechanical process adds a vaporization at 160 °C for 30 min of the raw material prior to its defibering. This treatment has a 95% yield. Finally, a chemo–thermomechanical process cooks the fibers with sodium hydroxide/antraquinone at 160 °C during 20 min. before defibering. This treatment has a 90% yield [22]. During these processes, the amount of lignin and extractives is reduced progressively. Thus, the processes ensure a higher amount of possible interactions but also reduce yield and increase dispersion difficulty. The literature shows how increasing the aggressiveness of the treatments increases the tensile and flexural strengths of the composites. Nonetheless, if this increase of the mechanical properties is confronted with the decreasing yields the advantage disappears [22]. Notwithstanding, in the case of ONPF even though being TMP, are recycled fibers that will be submitted to defibering. Thus, a 100% yield can be presumed. Anyhow, ONPF were previously defibered to obtain the TMP, and a new defibering will presumably shorten the fibers. Figure 5 shows the morphology of the ONPF extracted from a composite with a 30% *w*/*w* content.

These fibers showed a mean length of 266 µm and a weighted length of 665 µm. The highest amount of fibers was found in the range from 70 to 490 µm. In these cases, the use of weighted lengths is usual, to prevent misevaluations of the long fiber contributions. The mean weighted lengths for the 40% and 50% *w*/*w* ONPF reinforced composites were 526 and 416 µm, respectively [8,11]. SGW MP, extracted from 30%, 40% and 50% *w*/*w* reinforced PP composite returned a mean weighted length of 698, 670 and 549 µm, respectively [30]. These noticeable higher mean lengths of the SGW fibers can explain the higher flexural strengths.

In any case, a composite material must fulfill market requirement to be a reliable alternative to commodity materials like glass fiber reinforced PP composites. Figure 6 compares the flexural strength of ONPF and glass fiber (GF) reinforced PP composites. The flexural strength of GF-based composites was obtained from the literature and the authors only found values for 20% and 30% *w*/*w* GF composites as reinforcement of the same PP matrix [27].

Despite the use or not of coupling agents, GF-based composites showed higher flexural strengths, at the same reinforcement content, than ONPF-based composites. This was expected due to the high intrinsic properties of GF. Nonetheless, the coupled ONPF-based composites at 40% and 50% *w*/*w* contents showed flexural strengths comparable to the uncoupled GF-based composites. It is also worth noting that the flexural strength of the 20% *w*/*w* GF composite was only 8% higher than the 50% *w*/*w* ONPF composite.

From an environmental point of view, the higher content of ONPF needed to obtain competitive flexural strengths can be seen as an advantage. The higher is the amount of reinforcement the lesser is the PP content, being ONPF a renewable resource and PP a non-renewable oil-based material.

### 3.3. Analysis of the Contribution of ONPF and its Intrinsic Flexural Strength

The flexural strength of a short fiber semi-aligned composite is mainly impacted by the phases contents, its nature and its mechanical properties, the morphology and dispersion of the reinforcements, the strength of the interphase and the mean orientation of the reinforcements. Previously, the morphology of the reinforcements, the nature and contents of the phases and the strength of the interphase have been discussed. On the other hand, the mean orientation of the fibers is highly impacted by the geometry of the injection mold and the process parameters. Thus, the unknown is the intrinsic flexural strength of ONPF. This intrinsic flexural strength can be obtained experimentally or by using micromechanics models. The experimental acquisition is especially difficult in the case of natural fibers due to its inherent variability [35]. Additionally, some authors object that the intrinsic properties for the same fiber inside or outside a composite can change noticeably [36]. Thus, the authors opted for the use of micromechanics models.

One of the most and simple models used to compute the contribution of the phases to the flexural strength of a composite is a modified rule of mixture (mRoM). One possible enunciation of the mRoM for the flexural strength of a semi-aligned short fiber reinforced composite is:(2)$$\sigma_{f}^{C}=f_{c}\times \sigma_{f}^{F}\cdot V^{F}+\left(1-V^{F}\right)\times \sigma_{f}^{m*},$$
where *σ_f_^C^*, *σ_f_^F^* and *σ_f_^m*^* are the strengths of the composite the fiber and the matrix at the strain at break of the composite, respectively. *V^F^* is the volume fraction of reinforcement, and *f_c_* is a coupling factor, with values between zero and one, corresponding to the strength of the interphase, the mean orientation of the fibers and the morphology of the reinforcements.

In its current shape, the mRoM has two unknowns, *σ_f_^F^* and *f_c_*. For this reason, some authors propose reordering the mRoM keeping the unknowns on the left side:(3)$$\sigma_{f}^{F}\times f_{c}\;=\frac{\left(1-V^{F}\right)\times \sigma_{f}^{m*}-\sigma_{f}^{C}}{V^{F}}.$$

Equation (3) is a line that depends on *V^F^*. In the equation, *σ_f_^F^ × f_c_* corresponds to the contribution of the reinforcements to *σ_f_^C^*. Therefore, the contribution of the fibers will increase with the percentage of reinforcement. Then, if a regression line passing thorough the origin and the contributions at different reinforcement contends is computed, the slope of the line will be called fiber flexural strength factor (FFSF). Figure 7 presents the computed points and presents the equation of the regression line. The figure also shows the FTSF with values from the literature [8,11].

The FFSF for ONPF in a PP-based composite was found to be 155 (orange line). The figure shows how the contribution of the ONPF was higher for the 20% and 30% *w*/*w* content composites (blue line). In fact, the slope of the regression line corresponding to these two contributions was 212, noticeably higher than the mean one. The contribution slightly deviated to lower contributions for higher ONPF contents. This behavior was already present in the contributions to the tensile strength, but at a lesser scale. The diminution can be caused by the tougher dispersion of the reinforcements when its percentage increases, or its morphological changes. This value has meaning when it is compared with others. Figure 8 shows the FFSF for different composites, recovered from the literature [6,8,11,22,30,35,36,37].

The FFSF of ONPF/PP was noticeably lower that GF/PP composites, despite the use of coupling agents or not. GF shows strengthening capabilities 3.1 and 4.1 times higher than ONPF when reinforcing the same matrix. The difference is mainly due to the intrinsic properties of GF, much higher than those of natural fibers, and to the regularity of a man-made reinforcement [38,39]. Regarding the rest of the composites, the value of ONPF was found to be inferior to SGW, despite the matrix. A higher strengthening ability was expected from a TMP in front of a MP, but as above commented, ONPF is a TMP that has been summited to a mechanical treatment, with all the involved morphologic changes. Nonetheless, Figure 7 shows initial contributions in the order of 221, higher that SGW-based composites. More research is needed to found the causes of the diminution of the contributions of ONPF for high reinforcement contents.

Regarding the intrinsic flexural strength of ONPF. The literature offers models that allow its computation. One of these methods explodes a relation between FTSF and FFSF [29,34,35]:(4)$$\sigma_{f}^{F}=\frac{FFSF}{FTST}\times \sigma_{t}^{F}.$$

The ratio between FFSF and FTSF was 1.55 for all the values and 2.14 for the 20% and 30% *w*/*w* composites, respectively. The literature established that the intrinsic tensile strength of ONPF was 514 MPa [8,11]. Thus, the mean intrinsic flexural strength is 771.3 MPa for all the values and 1100.5 MPa for the 20% and 30% *w*/*w* composites, respectively. The literature situates the intrinsic flexural strength of SGW at 1095 or 888 MPa as PP or PA11 reinforcement, respectively. The literature shows how the chemical family of the matrices impacts the value of the intrinsic strength of the reinforcements [40]. Nonetheless, the differences between ONPF and SGW were higher than expected. If all ONPF contents are included, the intrinsic flexural strength of ONPF was 30% lower than SGW. Nonetheless, the intrinsic flexural strength computed for 20% and 30% *w*/*w* composites was virtually the same as SGW. Thus, it was identified as possibility to increase the flexural strength of ONPF-based composites at reinforcement contents higher than 30% *w*/*w*.

## 4. Conclusions

Composites of old newspaper fibers reinforced polypropylene were formulated, mixed and tested under three-point bending conditions. Two batches of materials were prepared, one without any coupling agent and the other with a 6% *w*/*w* of polypropylene functionalized with maleic anhydride. The reinforcing fibers were suspended in diglyme to avoid the creation of bonds between the hydroxyl groups in its surface and ensure a proper dispersion.

Uncoupled composites showed an initial increase of the flexural strength, but this initial increase stopped for higher reinforcement contents. The cause was attributed to a weak interphase due to the chemical incompatibility between the hydrophilic reinforcement and the hydrophobic matrix.

Coupled composites showed a quasi-linear increase of its flexural strength against reinforcement contents. Micrographs showed a proper wetting of the reinforcements without any observable void. The cumulative increase of the flexural strength decreased for reinforcement contents higher than 30% *w*/*w*.

Compared with a mechanical pulp reinforced polypropylene, ONPF showed similar flexural strengths at low reinforcement contents and lower values at higher reinforcement contest. A study of the morphology of the fibers showed that ONPF returned lower lengths than the mechanical fibers, hindering its potential strengthening capabilities. The shortening of the fibers was attributed to the fact that ONPF suffered two treatments, a thermos–mechanical treatment to obtain newspaper, and a mechanical treatment to defiber newspapers. On the other hand, the diminution of the flexural strength can be attributed to the increasing difficulty to disperse ONPF when its percentage increased.

Compared with commercial glass fiber reinforced composites (coupled and uncoupled), ONPF-based composites showed similar flexural strength than the uncoupled GF-based composites, but at higher ONPF contents. The flexural strength of the 30% *w*/*w* GF coupled composite was unreachable with ONPF-based composites.

The contribution of the ONPF was similar to SGW for the composites with contents up to 30% *w*/*w*. Higher ONPF contents returned contributions noticeably lower. The causes were attributed to the morphology of the recycled fibers, a worst dispersion or the creation bonds between ONPF. More research is needed to point out the cause. The intrinsic flexural strength of ONPF was lower than other natural fibers when the full spectrum of reinforcement percentages was considered. Nonetheless, when the 20% and 30% *w*/*w* ONPF composites were used to compute the intrinsic flexural strength, its value was almost the same to the obtained for SGW.

## Figures and Tables

**Figure 1 polymers-11-01244-f001:**
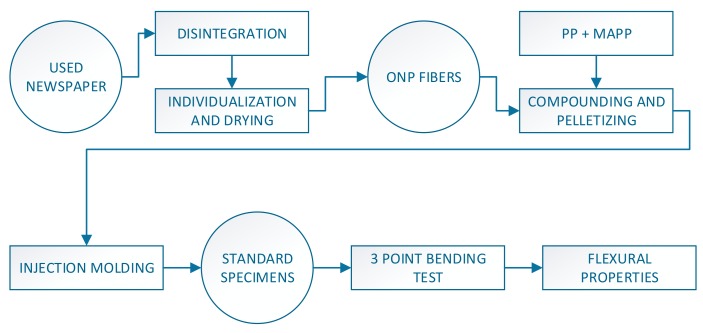
Flowchart of the research, from the raw materials to the experimental data.

**Figure 2 polymers-11-01244-f002:**
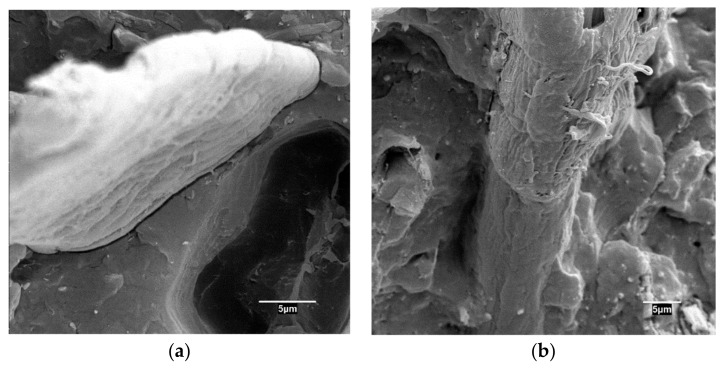
Micrographs of old newspaper fibers (ONPF) reinforced polypropylene (PP) composites, uncoupled (**a**) and coupled with a 6% *w*/*w* polypropylene functionalized with maleic anhydride (MAPP) (**b**).

**Figure 3 polymers-11-01244-f003:**
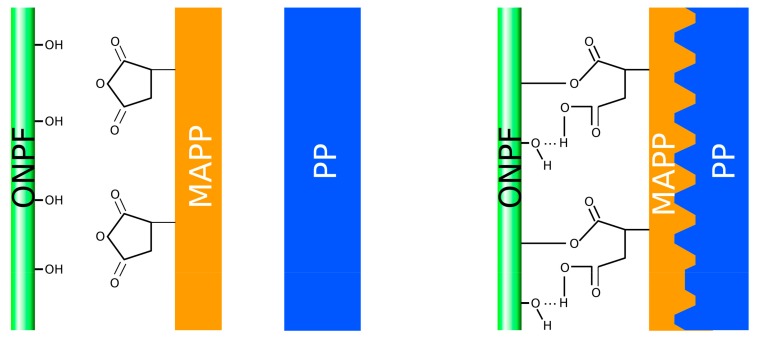
Schematic interactions between ONPF, PP and MAPP in the composite interphase.

**Figure 4 polymers-11-01244-f004:**
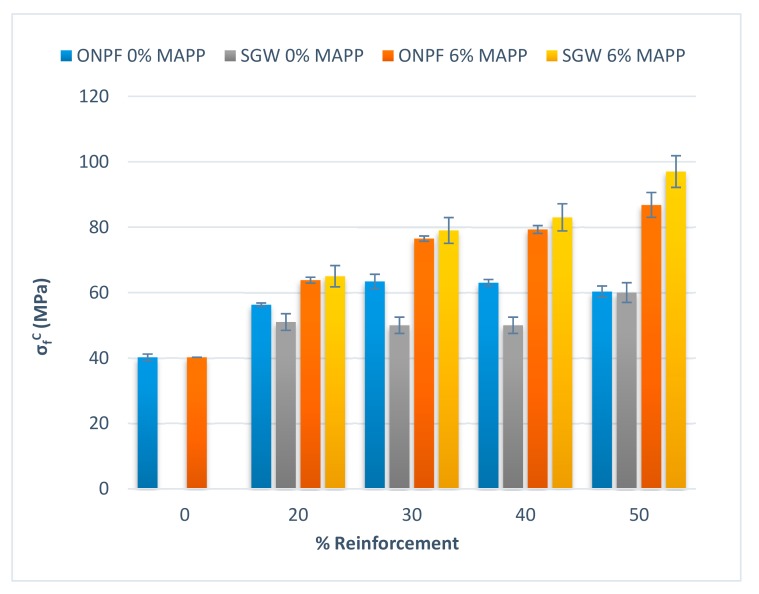
Comparison of the flexural strengths of coupled and uncoupled ONPF and SGW reinforced PP composites.

**Figure 5 polymers-11-01244-f005:**
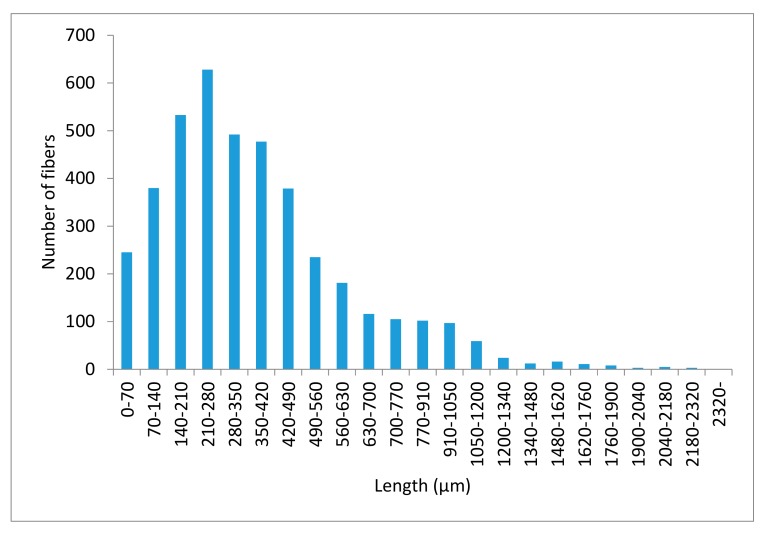
Fiber length distribution of the ONPF extracted from a 30% *w*/*w* composite.

**Figure 6 polymers-11-01244-f006:**
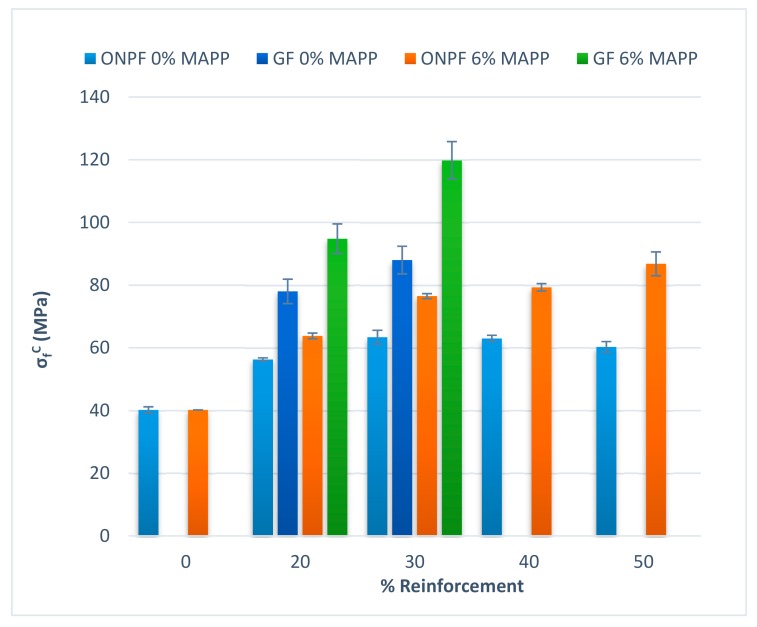
Comparison of the flexural strengths of coupled and uncoupled ONPF and GF reinforced PP composites.

**Figure 7 polymers-11-01244-f007:**
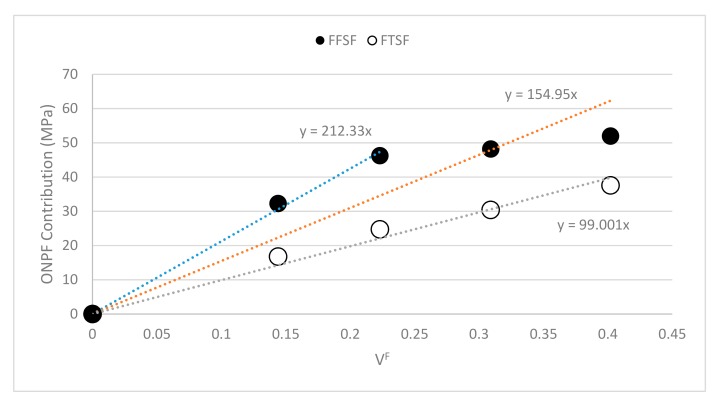
Fiber flexural strength factor and fiber tensile strength factor for the ONPF reinforced PP composites.

**Figure 8 polymers-11-01244-f008:**
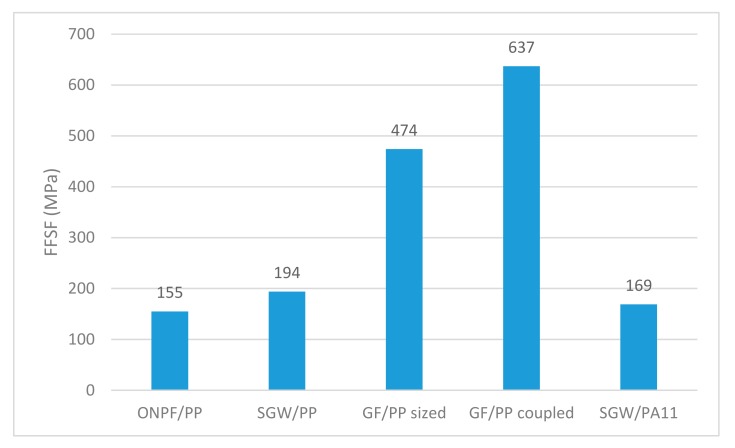
Fiber flexural strength factor of PP and PA11-based composites, reinforced with different fibers.

**Table 1 polymers-11-01244-t001:** Flexural properties of the old newspaper fibers (ONPF) reinforced Polypropylene (PP) composites against reinforcement and coupling agent contents.

Composite	*V*^F^ (*v*/*v*)	*σ*_f_^C^ (MPa)	*E*_f_^C^ (GPa)	*ε*_f_^C^ (%)	*σ*_f_^m*^ (MPa)
PP	0	40.2 ± 0.6	1.1 ± 0.12	6.5 ± 0.25	40.2 ± 0.6
20%ONPF–0%MAPP	0.144	53.6 ± 1.0	2.1 ± 0.16	3.7 ± 0.16	53.6 ± 1.0
30%ONPF–0%MAPP	0.223	63.4 ± 0.5	2.7 ± 0.22	2.7 ± 0.11	63.4 ± 0.5
40%ONPF–0%MAPP	0.309	63.0 ± 1.8	3.1 ± 0.19	2.6 ± 0.13	63.0 ± 1.8
50%ONPF–0%MAPP	0.402	60.3 ± 1.7	4.0 ± 0.23	1.6 ± 0.12	60.3 ± 1.7
20%ONPF–6%MAPP	0.144	63.8 ± 0.9	2.2 ± 0.14	4.2 ± 0.18	63.8 ± 0.9
30%ONPF–6%MAPP	0.223	76.5 ± 0.8	2.5 ± 0.18	4.0 ± 0.15	76.5 ± 0.8
40%ONPF–6%MAPP	0.309	79.3 ± 1.2	3.2 ± 0.24	2.6 ± 0.13	79.3 ± 1.2
50%ONPF–6%MAPP	0.402	86.8 ± 2.3	4.1 ± 0.21	2.5 ± 0.14	86.8 ± 2.3

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
