# Peer review of "Flexural Properties and Mean Intrinsic Flexural Strength of Old Newspaper Reinforced Polypropylene Composites"

_polymers, 2019, doi:10.3390/polym11081244_

Round 1
Reviewer 1 Report
This a very sound study of the use of recycled newspapers to reinforce polyproylene. The newspaper was disintegrated to free the component wood fibres, and these were used with and without coupling agent. This led to subtle differences in behaviour of the filled polypropylene. However, the presence of the newspaper fibres was generally beneficial and increased the flexural modulus of the resulting composites.
Overall, this work merits publication. However, it needs editing for quality of English. Also, the proper term for the class of polymer represented by polypropylene is polyolefin, not polyolephin as used by the authors.
Author Response
Answer to the reviewer 1
The authors thank the reviewer for his kind revision.
The authors have changed Polyolephin to polyolefin and beg their pardon to the reviewer.
The authors have also reviewed the English language.
Reviewer 2 Report
This is an interesting and important paper that presents the effect of Newspaper fiber and coupling agent on PP. Column 1 in Table 1 should be more descriptive like in Figure 4: 20%ONPF-6%MAAP, for example. The length axis in Figure 5 should be corrected. The first bar corresponds to 0-70, not to 0, etc.
Author Response
Answer to the reviewer 2
The authors thank the reviewer for his kind revision.
The authors agree with the reviewer and have changed the first column of table 1.
The authors have also changed the x-axis on figure 5. The reviewer is right; we thank him for his attention to detail.
The authors have also reviewed the English language.